# Agonist Treatment for Opioid Dependence Syndrome: The Impact of Current Understanding upon Recommendations for Policy Initiatives

**DOI:** 10.3390/ijerph181910155

**Published:** 2021-09-27

**Authors:** Cheryl Dickson, Valérie Junod, René Stamm, Emilien Jeannot, Robert Hämmig, Willem Scholten, Olivier Simon

**Affiliations:** 1Addiction Medicine, Lausanne University Hospital (CHUV), 1004 Lausanne, Switzerland; emilien.jeannot@unige.ch (E.J.); olivier.simon@chuv.ch (O.S.); 2Faculty of Law, University of Geneva, 1211 Genève, Switzerland; valerie.junod@unil.ch; 3Faculty of Business and Economics, University of Lausanne, 1011 Lausanne, Switzerland; 4Independant Counsultant, 3012 Bern, Switzerland; rene.stamm@hotmail.com; 5Faculty of Medicine, Institute of Global Health, Chemin de Mines 9, 1202 Geneva, Switzerland; 6Dr. Robert Ltd. on behalf of the Swiss Society of Addiction Medicine, 3000 Bern, Switzerland; robert.haemmig@hin.ch; 7Willem Scholten Consultancy, NL-3411 AD Lopik, The Netherlands

**Keywords:** opioid dependence syndrome, opioid agonist treatment, OAT, harm reduction, legislation, policy

## Abstract

The provision of opioid agonist treatments (OATs), as a standard approach towards opioid dependence syndrome, differs widely between countries. In response to access disparities, in 2014, the Council of Europe’s Pompidou Group first brought together an expert group on framework conditions for the treatment of opioid dependence. The group used a Delphi approach to structure their discussions and develop guiding principles for the modernisation of OAT regulations and legislation. The expert group identified some 60 guiding principles, which were then the subject of wide public consultation. Endorsed by Pompidou Group member states, the final report identified four key recommendations: (1) Prescription and delivery without prior authorisation schemes; (2) Effective removal of financial barriers to access to care; (3) Coordination and follow-up by a national consultative body; and (4) Neutral, precise and respectful terminology. During meetings, the expert group hypothesised that inequalities in OAT access are likely to be linked to underlying rationales which in theory are contradictory, but in practice co-exist within the different political frameworks. The present article considers the perceived influence upon different regulatory frameworks. Discussion is centred around the potential impact of underlying rationales upon the effective implementation of a modernised framework.

## 1. Introduction

The prescription of morphine and other opioids for the long-term treatment of Opioid Dependence Syndrome (ODS) has been practiced for more than 150 years. From the 1880s onwards it was accepted as a standard medical approach in some countries [1]. However, its practice rapidly declined in most countries during the 1920s, following the introduction of a prohibitionist approach in the United States of America [2]. Yet, the United Kingdom continued such treatment until the 1980s and developed a standardised model for prescription [3]. The late 1960s saw a change in perspective in the US, largely brought about by the work of Dole and Nyswander, which led to the re-introduction under the auspices of “methadone maintenance” [4,5]. Treatment availability was limited in the US throughout the 1970s, due to Nixon’s administration and its implementation of Prior Authorisation Schemes (PAS) by the Drug Enforcement Administration [6]. In practice, such highly restrictive schemes limited access to medicines and care. Implementation of PAS in the US was imitated elsewhere, and thus methadone maintenance became a theoretically possible, yet highly regulated intervention, in many countries, including Switzerland [7].

In the 1980s, the Acquired Immunodeficiency Syndrome (AIDS) crisis brought about a new focus on the prescription of methadone since it could reduce the risks of infection among those injecting heroin or cocaine. As a public health response to the crisis, the concept of harm reduction (HR) pragmatically emerged [8]. From this perspective, opioid prescription came to be seen as a “substitution” treatment and a potential component of HR. Nonetheless, both the medical community and policymakers were divided over its use, giving rise to wide access disparities [9]. 

In 2005, in recognition of their medical benefits as a pharmaceutical treatment, both methadone and buprenorphine were added to the World Health Organization (WHO) Model List of Essential Medicines, giving them the status of “minimum medicine need(ed] for a basic healthcare system” [10]. Ten years later, in 2015 the United Nations (UN) General Assembly identified, among its Sustainable Development Goals to reduce Human Immunodeficiency Virus (HIV) and Hepatitis C virus, access to treatment for substance dependence and access to essential medicines. In 2016 the European Court of Human Rights (ECHR) issued a landmark ruling on State obligations regarding Opioid Agonist Treatment (OAT) provision in detention settings. In this landmark case (Wenner V. Germany), the defendant was refused the right to continue his current methadone treatment following entry into prison. The court relied on Article 3 of the European Convention of Human Rights, relating to the prohibition of torture, and inhumane or degrading treatment and ruled that any member state refusing access to OAT must prove that an alternative medical approach would be as effective, for the person in question. They also deemed that this proof must be established by independent medical opinion, rather than reliant upon State decision [11]. However, in another more recent case, (Abdyusheva and others V. Russia) the court concluded that Russia could decide which medical approach to favour and which to forbid, including decisions over whether to provide a “conventional” abstinence-based approach or OAT, as long as the patients were offered a form of “medical treatment”. The wide margin of appreciation given to Russia, in this case, was seen as a missed opportunity to set the golden standard of effective and ethical medical treatment for opioid dependence syndrome [12].

However, despite the recognition of OAT treatment benefits [13] by certain authorities and organisations, perceptions of their nature and clinical utility vary and their availability remains disparate between European countries, and beyond. Currently, access to methadone is possible in primary care settings, with access to methadone being possible in primary care settings in Ireland, Romania, Slovenia and Spain, access to buprenorphine is possible in The Balearic Islands, Croatia, The Czech Republic and Latvia, both medicines are accessible in Austria, Belgium, Corsica, Denmark, France, Germany, Italy, Luxembourg, Northern Ireland, Norway, Portugal, Sardinia, Sicily, Switzerland, The United Kingdom and neither are available in Bulgaria, Crete, Estonia, Finland, Greece, Hungary, Lithuania, Malta, The Netherlands, Poland and Slovakia [8]. Variations in availability are also seen for places of detention [14]. In addition, financial coverage rates for people who inject opioids exhibit wide variations [9]. Access and availability are reported to be influenced by the implemented framework conditions, with “flexible” structures, a low threshold for access to care and ongoing availability of opioids, being identified as beneficial to the provision of OATs [15]. In response to such inequalities, in 2014, the Pompidou Group (an intergovernmental organisation affiliated to the Council of Europe) mandated an expert group to consider international efforts to regulate OATs. The group was composed of 18 experts from various European, Middle Eastern and Northern African (MENA)-countries (see [16] p.98 for a detailed list of experts and their backgrounds). At its initial meeting, the wide disparities in the coverage rates and their connection to different underlying frameworks were highlighted. Perspectives on the nature and benefits of OATs were felt to strongly influence such variations. In light of these findings, the need to develop guiding principles to review and modernise regulations was underlined.


*Scope of the Present Article*


The current article will present the hypothesis on underlying frameworks for OAT that was developed during Pompidou Group’s expert group meetings. In order to provide a context to their perspective, an overview of their work on the guiding principles will first be provided. The expert group’s hypothesis will then be presented, including a description of two different underlying rationales for the provision of OATs. The discussion will be centred around the impact of these perspectives on the provision of OATs and potential barriers to their provision.


*The Pompidou Group’s Approach*


Following a proposal from the Swiss Federal Office of Public Health, the Pompidou Group mandated an international expert group to review OAT regulations and legislation. Legal and healthcare experts from various European and MENA countries took part in four formal meetings and a Delphi survey carried out between August 2014 and May 2017. The varying framework conditions for OAT and underlying perspectives on the nature of and need for access to OAT were discussed from the first meeting, onwards. In order to develop the guiding principles, a Delphi Survey questionnaire was then developed, which drew on the opinions and experience of professionals both inside and outside of the expert group. The survey was administered in three waves (please see [16] pp. 39–41 for details of participants for each wave of the Delphi survey). Subsequent meetings focused on the development of guiding principles and the writing of a report. Through this process, 62 guiding principles were identified, which then underwent a period of public consultation. From these 62 principles, 4 strategic recommendations were highlighted [16].


*The Structure and Content of the Guiding Principles and Key Recommendations*


Following their hypotheses on the underlying rationales, and influence upon current access disparities, the expert group went on to consider the fundamental right to treatment in international law, which implies the need to review and modernise existing frameworks for OAT. The Pompidou Group expert group then worked to develop guiding principles for OAT regulations and legislation. The above-mentioned 62 guiding principles and 4 strategic recommendations were identified to assist countries in reviewing and modernising OAT frameworks. The general structure of the principles is presented in Table 1. below and the full report [16] is available on the Pompidou Group website in English, French, Italian, German, Spanish, Czech, Polish, Russian and Arabic.


*Prescription and Delivery of OAT without PAS*


OATs were recognised as the best long-term treatment for opioid dependence. As a consequence, their administration should be part of the basic responsibility of healthcare professionals, in particular physicians and pharmacists. The PASs should not be used to oversee the administration of Opioid Agonist Medicines (OAMs), because they go hand-in-hand with lengthy and complicated administration processes. Such schemes act as a barrier to care and access to essential treatment. The Pompidou Group, therefore, recommends the elimination of PAS: responsibility should be placed with healthcare professionals and regular supervising bodies, in order to guarantee improved availability, accessibility, acceptability and quality of treatments.


*Removal of Financial Barriers*


People with opioid dependence syndrome can be particularly vulnerable and difficult to reach. In addition to their addictive behaviours, they may experience other comorbid disorders, social marginalisation and demonstrate high-risk behaviours. OAT accompanied by psychosocial counselling or somatic care is recognised as essential to their treatment. Yet, such care can be beyond the budgets of those who need it most. Additionally, lengthy, complex and stigmatising steps to access funding can pose as a major deterrent. To suppress financial barriers, the guiding principles state that treatment, including first visits, OAT prescription and follow-up, should be provided at no cost to the individual. To make this possible, the implementation of a special funding scheme may be necessary, which differs from general healthcare financing. Such schemes are justified on public health grounds and can be offset by savings on direct costs (such as those of social and legal services), indirect costs (such as loss in productivity for individuals and their support network), and intangible costs (such as loss of quality of life).


*An Independent National Consultative Body*


In some countries, OAT is monitored by government agencies. However, depending on their nature, these agencies can be severely compromised by conflicts of interest. In other countries, there is less focus on feedback and monitoring, with no specialised organisation. The introduction or replacement of agencies by an independent national consultative body would enable objective monitoring and feedback on the implementation of standardised treatment processes. The expert group proposes that a panel of multidisciplinary and multi-agency representatives should be brought together for this task, including people who use opioids non-medically. There should be a focus on independence and transparency regarding its budget and panel composition (notably, declarations of interest), which should be made available through regular public reports. The body should promote scientific research and act as an interface between on-the-ground experiences and knowledge derived from such research.


*Neutral, Precise and Respectful Terminology*


The words used to describe individuals and their behaviours can in themselves be negative and stigmatising. Their influence upon public perceptions should be recognised; in order to avoid misunderstanding, terminology that is neutral, precise and respectful should be adopted.

A number of negative and ambiguous terms are currently in regular use, which may cause confusion over the disease and treatment. In particular, terms such as “user” imply choice, place blame, and identify the person as defined first and foremost by his behaviour whereas “person who consumes opioids” is a person-first alternative. Moreover, the term “substitution” was highlighted as particularly ambiguous and not in keeping with scientific knowledge, whereas “treatment” reflects a more accurate description. Efforts to adopt neutral, precise and respectful language by professionals, administration and policymakers would influence the understanding of OATs at a community level. A few examples [17,18] can be found in Table 2. below. As terminology preferences will vary between languages and localities, the expert panel encourages consideration of this issue by different countries and regions.


*Underlying Rationales*


Discussions from the expert group meetings were shaped both by professionals’ observations and issues raised within the Delphi survey. These conversations led to the consensus that the existing structural frameworks of the different countries vary in the degree of flexibility or restriction that they allow. These approaches were conceptualised as being linked to a set of beliefs about the nature of OATs and their use as a medical treatment. The following two distinct theoretical perspectives were hypothesised:


*Rationale a*


This approach rests on the understanding of OAT as a substitute product to replace street drugs, with the aim of providing a safer supply. From this perspective, OAT is seen as a harm reduction measure, rather than medical treatment per se, according to a definition of harm reduction as *aiming to reduce harm to the general population* [19]. This approach is aligned to general human rights such as the right to life, social rights and freedom from discrimination. The State does not consider treatment to be mandatory under a fundamental right to access to care. As such, some jurisdictions may choose to provide OAT for certain individuals, and in certain settings (healthcare or more rarely, health services in places of detention). Provision of OATs from this viewpoint is seen as the special authorisation of something that is not otherwise delivered. To oversee its administration, PAS are typically used and are often monitored by ad hoc supervisory instances. The structure provided by PAS can present a barrier to access in some circumstances. Fully licensed physicians must obtain permission from the administration or from a medical government agency prior to administration and physicians and pharmacists are considered to be administering treatment as an extraordinary part of their role. Delivery can be withdrawn if deemed necessary due to criteria defined by the public authorities and influenced by a changing political context [15,20]. Such an approach means that access to OAT can be lengthy, limited or not currently available in some countries [9].


*Rationale b*


Under rationale b, OAT is seen as an essential treatment option, in keeping with its current status on the WHO Model List of Essential Medicines [10]. Opioid Agonist Medicines (OAMs) are recognised to have very different effects from opioids used for non-medical hedonic purposes; they stabilise the emotional state and reduce the subjective reinforcing effects causing dependence. Unambiguous results of clinical trials have confirmed the effectiveness of OAT: it reduces risks of infection, accidental death and suicide associated with opioid consumption. Access to OAT implements the fundamental right to access to healthcare. From this perspective, there is a clear State obligation to provide treatment, implying also a straightforward and timely pathway. To achieve this obligation, fully licensed physicians and pharmacists are responsible for its provision, and the usual supervisory bodies (medical agencies and regular professional bodies) ensure that best practices are being followed. Under this rationale, PAS are avoided as they would induce barriers and interfere with the principle of therapeutic freedom. Continuous research and innovation are encouraged, in line with current pharmaceutical and medical research standards to inform OAT administration. Rationale b is a person-centred approach, placing an emphasis upon the needs and autonomous choices of the individual.

Consensus of the Delphi Enquiry

The Delphi enquiry that served to develop the guiding principles also led to further discussion by the expert group on the two conceptualised perspectives. The group concluded that whilst subscribing to different underlying philosophies, both approaches appear to co-exist within certain political frameworks. However, there was a general consensus from the group towards the benefits of perspective b. Particularly, it was felt that providing ordinary controls for OAT would enable increased access and timely treatment, thus fulfilling the fundamental human right for access to healthcare. As OAT medicines methadone and buprenorphine are on the WHO Model List of Essential Medicines, The expert group deemed that all International Covenant on Economic, Social and Cultural Rights (ICESCR) signatory countries have a responsibility to make these medicines available and therefore a framework to facilitate supply is essential.

## 2. Discussion and Conclusions

Current clinical research shows the beneficial effects of OATs, and on this basis, they are recognised by the Pompidou Group as the standard treatment for opioid dependence syndrome. Yet, the general legal framework existing in different countries appears to influence the current structure of the specific framework for access to OATs. In particular, jurisdictions perceiving OAT as a treatment, rather than a safer supply (substitute) product, are believed to prioritise practices to ensure its availability. Elsewhere, the perception of OATs as a supplementary measure in the interest of the general population is felt to be linked to more restrictive administrative practices. The approach taken by different countries can change, over time but a recognition of the essential nature of OATs will be necessary to facilitate the adoption of a framework that favours treatment access.

Even where treatment practices are already prioritised, there is a risk that by making OATs subject to ordinary administrative and supervision practices, this treatment option could become less of a priority over time. Such circumstances might lead to future misunderstanding, underfunding or even the risk of a public health crisis. There is therefore a role for political follow-up mechanisms to maintain a high-profile presence and drive these issues forwards, in order to avoid unnecessary barriers to OAT. Funding is one potential issue and it will be particularly important that OATs are financed by government schemes, and not at the cost of individuals to ensure equity of access.

Furthermore, an independent national consultative body was proposed in order to objectively monitor and feedback on treatment processes. Such bodies should enable monitoring and coordination that is free from political, ideological or regulatory capture. Centralising decision-making need not unduly compromise political accountability/regional autonomy but it will be important to ensure that structures are in place to safeguard against these eventualities.

As mentioned above, the successful implementation of an adapted and modernised framework for access to OATs relies on a clear understanding of the essential nature of agonist medicines. Common misconceptions over this treatment are likely to impede both political discussion and public support for a move in this direction. Parallels can be seen here with the use of benzodiazepine, stimulants or medicinal cannabinoids. In these instances, recognition of the medical need for such treatment is key to its global acceptance and widened availability. In addition, concerns were raised about adopting a permissive approach to OAT delivery. Specifically, it is feared that dependence will be induced or increased by OAT availability, and/or the illegal market could benefit from OAM leakage. Whilst leakage and increased dependence are real possibilities, the alternative (a less permissive approach to OAT provision) was recognised as severely detrimental to those with OAT dependence and thus raises a significant public health issue. To facilitate the proper understanding of OAT, political debate and public awareness campaigns could play a significant role. However, like the response towards the HIV crisis in the 1980s, it is essential that the subject matter receives an assured place on the political agenda, sustained interest and long-term guaranteed funding to ensure ongoing provision. A particular research focus from the legal academic field should highlight the conflict of rights between the legislation of therapeutic products and laws relating to controlled substances, which presents a current gap in the literature. Pompidou group guiding principles are a starting point in this direction, but it is important that they are understood as a tool to encourage review and stimulate legislative discussion.

## Figures and Tables

**Table 1 ijerph-18-10155-t001:** General structure of the Pompidou Group’s Guiding Principles.

Part	Section	Main Messages
**I-Definitions** **and objectives**	1 - Definitions 2 - Objectives of the principles 3 - Objectives of Opioid Agonist Medicines (OAMs)	Primary objectives of OAMs centred on the person and the fundamental right to access essential medicines. Opioid Agonist Treatment (OAT) scientifically recognised for Opioid Dependence Syndrome (ODS) as well as a risk and harm reduction measure.
**II-Right to OAMs and related care**	4 - Fundamental right to healthcare 5 - Non-discriminatory access 6 - Free and informed consent 7 - No discrimination due to the simple fact of receiving OAMs 8 - Continuity of care	Respect for the principle of non-discrimination de jure and de facto justifying monitoring and ad hoc measures. Compliance with the principle of equivalence of care. Right of access to the treatment for civil minors. Guaranteed continuity of the medicine even in the case of an impasse in the provider–person treatment relationship. No delay to the start of treatment once the indication was established.
**III-Role of the professionals**	9 - Indication, prescription, dispensation, coordination 10 - Training of Physicians 11 - Training of Pharmacists 12 - Supervision	Competence to implement OATs expected from all physicians and pharmacists at the end of basic training; Right to prescribe granted to any physician. First-line monitoring of healthcare professionals by professional bodies (professional or disciplinary law); Importance of support measures alternative to sanctions (e.g., mentoring, group exchanges, supervision/intervision.
**IV-Role of the authorities**	13 - Availability and quality of OAMs 14 - Proportionality of the framework 15 - Financing and remuneration 16 - Training and research 17 - Monitoring and indicators	Authorisation of medicines and pharmacovigilance by the medicine agency. Summary of the Product Characteristics/Product Information providing basic information according to the standards applicable to any medicine. Abolition of Prior Authorisation Schemes (PAS). Possibility of declarative systems for the prevention of double prescriptions and epidemiological monitoring (if necessary). Specific mechanisms to remove financial barriers to treatment. Incentives for professionals guaranteeing effective availability of appropriately trained professionals. Guaranteed protection of personal data.
**V-National co-ordination and international collaboration**	18 - National consultative body 19 - International collaboration	Body integrating monitoring, professionals, users, state, parastatal, and private bodies to ensure the monitoring of regulatory revision efforts and their impact on healthcare systems. Standardisation of monitoring efforts, public reports. Financing and promotion of international guidelines rather than national guidelines.

Reproduced with permission from the Pompidou Group [16].

**Table 2 ijerph-18-10155-t002:** Examples of problematic terms and proposed alternatives.

Problematic term	Description	Alternative(s)
Addict	Not person-first language (reducing the person to one characteristic), pejorative and stigmatising in certain circumstances	Person with substance use disorder, or: person with dependence
Dependent, or: dependent person	Not person-first language (reducing the person to one characteristic)	A person with a substance use Disorder
Substitution therapy, or: Opioid Substitution Therapy (OST)	Misleading: gives the impression to politicians, civil servants and other lay people that this therapy is replacing “street drugs” with “state drugs” and therefore this language counteracts the availability of therapy	Therapy, Opioid agonist therapy (OAT), opioid agonist therapy for the treatment of substance use disorder; treatment
Higher risk group	Implies that the risk is contained within the group; can increase stigma and discrimination against the designated groups; membership of groups does not place individuals at risk, behaviours may.	Key populations; priority population; high-risk behaviour (e.g., sharing needles, condomless sex)

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
