# Peer review of "Agonist Treatment for Opioid Dependence Syndrome: The Impact of Current Understanding upon Recommendations for Policy Initiatives"

_ijerph, 2021, doi:10.3390/ijerph181910155_

Round 1

Reviewer 1 Report

This manuscript describes some of the work of the pompidou expert group to promote access to opioid agonist treatment (OAT) in European countries. It is well written, precise and distinguishes between higher and lower threshold approaches, namely OAT that is patient-centred and gives individuals using opioids easier treatment access by interaction with their physician, or OAT that is allocated more as an harm-reduction approach outside regular treatment and following a prior authorisation scheme (PAS). The group explains both models and favours the latter, which appears reasonable. The paper is interesting enough and well-balanced in language with a clear focus on legislation and differences between countries. 

The manuscript entitled “Agonist treatment for opioid dependence syndrome: the impact of current understanding upon recommendations for policy initiatives” describes the Pompidou expert group’s work with delineating the legal framework for recommendations to establish or improve best practice opioid agonist treatment (OAT) in Europe. The aim of the paper is to enhance the insight into the regulatory environment that is the basis for guidelines on common treatment models. The authors first give a brief overview on the history of how OAT is organized and then describe in detail two distinct approaches with benefits and drawbacks. These models are the basis to establish guiding principles for OAT in Europe and to describe briefly the work of the expert group between 2014 and ‘17. Special emphasize is put on “prior authorization schemes - PAS” which have introduced an extra bureaucratic hurdle and left OAT outside of standard medical care for several decades. The authors succeed in arguing for modern OAT based on decisions made by healthcare professionals instead of overarching administrations on regional or national level. They remind us of the inclusion of methadone and buprenorphine in the WHO’s list of essential medicine and point out that financial barriers for OAT patients need to be removed due to the target individuals’ vulnerable nature. The paper focuses on the existing legal frameworks for OAT as a medical treatment, which is handled quote differently in Europe. Political efforts to improve OAT access across Europe are still needed and the authors point out the role of the Pompidou group in giving clear advice on OAT program characteristics such as ease of access, sufficient funding and legal frameworks that secure its further development to effectively incorporate patient populations that are oftentimes hard-to-reach. I find the paper well written and clear in its argumentation, and think it is an important contribution from the legal field to further development of OAT in Europe. I have only a minor suggestion for improvement concerning the listing of the specific countries that offer either methadone or buprenorphine or both (cf. page 2, lines 84 to 87).

The only minor comment I have for improvement is that I would like to have the countries spelled out that have OAT with methadone, Buprenorhine, both or none (cf. page 2, lines 84 to 86). This could be solved as a table or preferably directly in the text.

Author Response

Thank you so much for your positive appraisal of this manuscript and for recognising its contribution to the legal field. 

Please find attached the additional information that you requested on the countries offering access OAT medicines  (lines 84 - 91). For your information, we have also carried out another spell check of the document before resubmitting.

Yours sincerely,

Cheryl Dickson

Reviewer 2 Report

General comments:

This viewpoint piece offers an interesting and important perspective on the ongoing debate surrounding accessibility of opioid agonists and the regulatory structure that conditions that access. It also offers an insightful look at the dueling philosophical frameworks that underlie much of the current approaches towards making these treatments available to people with substance abuse disorders. After laying out these opposing approaches, the authors make a case for a more permissive stance towards access, prescription, and use of opioid agonists. However, I think acknowledging and addressing some key problems with the general approach they lay out will strengthen the viewpoint. I have the following comments/suggestions for the authors:

  1. Authors describe the process by which the expert group generated four key recommendations to modify current regulatory structure/approaches. Although these recommendations are well described, the cumulative effect of all four would be a substantial increase in access to and use of opioid agonist for those with opioid dependence, as well as significant spillovers to the unaffected but vulnerable population through leakage. The key tradeoffs associated with such a seismic change are not adequately explored in this piece. Opioid dependence is already surging in Western nations and there is increasing concern that an unduly permissive approach to prescription and monitoring may be contributing to the epidemic. It is now acknowledged that permissive stances towards opioid treatment of chronic pain taken in nineties may have in some cases fostered and worsened dependence. I encourage authors to acknowledge these serious criticisms and attempt to meaningfully address them in their viewpoint.

  1. As authors note, there are significant variations in approaches taken by different countries, with some taking a more permissive approach that includes higher autonomy for physicians in prescribing and monitoring, limited pre-authorization, and lower thresholds to access. Exploring the impact of these variations could yield important answers to questions that need to be answered. Have these approaches yielded insights into these key tradeoffs? Is there a relationship between a more permissive approach and incidence of opioid use/dependence? What is the track record of existing harm reduction approaches in striking balance between adequate access to appropriate treatments and curbing potential gateways to higher recreational use? Would the same goals be achieved better with partial agonists rather that full agonists? Authors could provide a better general sense of the empirical literature on this issue in order to reassure the audience.

  1. Authors’ suggestion to set up national consultative bodies impervious to political pressures seems laudable on surface. Such bodies tend to follow the “disinterested expert” model of decision-making but have been criticized on grounds of insulation from political accountability and risk of ideological/regulatory capture. These criticisms can be addressed by pointing out that centralizing decision-making need not unduly compromise political accountability/regional autonomy if there are structures in place to safeguard against these eventualities.

  1. More information is needed regarding the process used to generate the recommendations. In particular, the process of selection of experts by the Pompidou group needs to be clarified. If (hypothetically) the selection were marred with ideological biases, the resulting consensus would be non-representative of the wider opinion and seriously misleading. The Delphi survey or a previously published link could be provided in appendix with the article to make the process more transparent.

  1. This reviewer is unconvinced that the term “substitution therapy” is less precise and needs to be replaced with “treatment”. Harm reduction approaches were grounded in a solid understanding of the dynamic underlying substance abuse: that unmonitored intake of opioids for recreational use resulted in serious compromises of health/safety. The understanding was that supervised use of agonists (partial or complete) supplied the same “relief” for addicted patients but in much more safe manner. In a subtle way, the term “treatment” is no more neutral than “substitution”: it implies that taking opioids for recreation is a neutral act in itself and completely removes any role of agency or moral judgment even in cases where it may be appropriate.

Author Response

-

This viewpoint piece offers an interesting and important perspective on the ongoing debate surrounding accessibility of opioid agonists and the regulatory structure that conditions that access. It also offers an insightful look at the dueling philosophical frameworks that underlie much of the current approaches towards making these treatments available to people with substance abuse disorders. After laying out these opposing approaches, the authors make a case for a more permissive stance towards access, prescription, and use of opioid agonists. However, I think acknowledging and addressing some key problems with the general approach they lay out will strengthen the viewpoint. I have the following comments/suggestions for the authors:

 Thank you for your comments and suggestions, which we have responded to below.

  1. Authors describe the process by which the expert group generated four key recommendations to modify current regulatory structure/approaches. Although these recommendations are well described, the cumulative effect of all four would be a substantial increase in access to and use of opioid agonist for those with opioid dependence, as well as significant spillovers to the unaffected but vulnerable population through leakage. The key tradeoffs associated with such a seismic change are not adequately explored in this piece. Opioid dependence is already surging in Western nations and there is increasing concern that an unduly permissive approach to prescription and monitoring may be contributing to the epidemic. It is now acknowledged that permissive stances towards opioid treatment of chronic pain taken in nineties may have in some cases fostered and worsened dependence. I encourage authors to acknowledge these serious criticisms and attempt to meaningfully address them in their viewpoint.

Our aim in this manuscript is to put forward a viewpoint based on the work of the expert group. This has led to 4 key recommendations, which are designed to facilitate progressive change and to gradually integrate (rather than oppose) the two underlying perspectives described in the manuscript. The idea is that gradual change can be initiated through one or other of the key recommendations, according to political opportunities within the different countries. The recommendations are designed to be interdependent and thus implementing one can lead onto implementing another.

Regarding the North American crisis, it is recognised that part of the problem is precisely the lack of access to care for the treatment of addiction syndrome and that an increase in treatment options is an essential part of an effective global approach (as per the Global Commission on Drug Policy`s report entitled Taking Control: Pathways to drug policies that work)- In order to manage the risks of  developing a framework, which itself poses a public health risk, the experts have insisted upon monitoring  (recommendation 3, section 17 of the guiding principles).

The expert group strongly felt that a less permissive approach to OAT provision would be at the extreme detriment of those with opioid dependence and a broader public health issue. Our article therefore presents this viewpoint, and whilst we briefly touch on the issue of dependence/leakage within the discussion section (which we recognise as a legitimate concern) we prefer not to diverge greatly from the expert group`s perspective.

  1. As authors note, there are significant variations in approaches taken by different countries, with some taking a less restrictive approach that includes higher autonomy for physicians in prescribing and monitoring, limited pre-authorization, and lower thresholds to access. Exploring the impact of these variations could yield important answers to questions that need to be answered. Have these approaches yielded insights into these key tradeoffs? Is there a relationship between a less restrictive approach and incidence of opioid use/dependence? What is the track record of existing harm reduction approaches in striking balance between adequate access to appropriate treatments and curbing potential gateways to higher recreational use? Would the same goals be achieved better with partial agonists rather that full agonists? Authors could provide a better general sense of the empirical literature on this issue in order to reassure the audience.

We are not aware of any existing evidence to support the argument that a permissive approach to OAT leads to increased recreational consumption. Neither have we witnessed this from working within the field. However, the absence of such evidence does not fully resolve this debate and therefore monitoring and evaluation of the evolving regulatory frameworks is highly important (as per recommendation 3 of the guiding principles: implementation of an independent national consultative body for monitoring).

  1. Authors’ suggestion to set up national consultative bodies impervious to political pressures seems laudable on surface. Such bodies tend to follow the “disinterested expert” model of decision-making but have been criticized on grounds of insulation from political accountability and risk of ideological/regulatory capture. These criticisms can be addressed by pointing out that centralizing decision-making need not unduly compromise political accountability/regional autonomy if there are structures in place to safeguard against these eventualities.

Thank you for raising this issue, which we have added to the discussion section.

  1. More information is needed regarding the process used to generate the recommendations. In particular, the process of selection of experts by the Pompidou group needs to be clarified. If (hypothetically) the selection were marred with ideological biases, the resulting consensus would be non-representative of the wider opinion and seriously misleading. The Delphi survey or a previously published link could be provided in appendix with the article to make the process more transparent.

Detailed information on the expert group panel, an overview of the Delphi process (including the various participants for each wave of the survey) and information on the wider public consultation can be found in the Guiding Principles for OAT Legislation and Regulations. We have therefore adapted the manuscript to refer the reader to appropriate sections of the report for this information.

  1. This reviewer is unconvinced that the term “substitution therapy” is less precise and needs to be replaced with “treatment”. Harm reduction approaches were grounded in a solid understanding of the dynamic underlying substance abuse: that unmonitored intake of opioids for recreational use resulted in serious compromises of health/safety. The understanding was that supervised use of agonists (partial or complete) supplied the same “relief” for addicted patients but in much more safe manner. In a subtle way, the term “treatment” is no more neutral than “substitution”: it implies that taking opioids for recreation is a neutral act in itself and completely removes any role of agency or moral judgment even in cases where it may be appropriate.

Thank you for sharing your thoughts on this issue. You are not alone in your perspective on this terminology. We do, however, feel that “substitution therapy” alludes to the idea of replacing a street drug by a legal alternative, which somewhat loses the focus on actual treatment. As the article was written in order to express a viewpoint on the matter, we prefer to retain our argument.

For you information, we have also carried out a spell check of the document before resubmitting.

Thank you, once again for the opportunity to respond to your comments and enhance our manuscript.

Reviewer 3 Report

In present form, the article cannot be accepted for publication.

The information existing in the analyzed manuscript is far too inconsistent for a review article.

The following important aspects are not detailed in connection with the presented topic: misconceptions regarding opioid addiction; balancing the need for adequate pain relief versus the treatment of opioid addiction; the current WHO recommendations in the field.

The text is difficult to read due to the improper use of some English terms and the way the sentences are organized, which causes a large part of the transmitted information to be diluted. Moreover, many of the abbreviations are not explained (AIDS - line 55; UM - line 64; HIV - lines 65, 267; ECHR - lines 66-67; MENA - line 111; ICESCR - line 238).

Discussions need to be substantially improved and detailed highlighting existing controversies, especially those related to differences in legislative regulations in various countries.

For documentation, the authors used a small number of bibliographic references and recent information in the field is missing.

Author Response

-

  1. The information existing in the analyzed manuscript is far too inconsistent for a review article.

Thank you for taking the time to review this article. It is intended as a viewpoint article, putting forward the authors` perspective and showcasing the work of the Pompidou Group`s expert group.

  1. The following important aspects are not detailed in connection with the presented topic: misconceptions regarding opioid addiction; balancing the need for adequate pain relief versus the treatment of opioid addiction; the current WHO recommendations in the field.

As per our point above, this viewpoint article presents the work of the Pompidou Group`s expert group in developing the guiding principles for OAT legislation and regulations. Whilst we touch on the issues of misconceptions over OATs, in the discussion section, we feel that adding the broader range of subjects associated subjects that you have suggested would significantly change the perspective that we are presenting in our article.

  1. The text is difficult to read due to the improper use of some English terms and the way the sentences are organized, which causes a large part of the transmitted information to be diluted. Moreover, many of the abbreviations are not explained (AIDS - line 55; UM - line 64; HIV - lines 65, 267; ECHR - lines 66-67; MENA - line 111; ICESCR - line 238).

We wonder whether some of the wording sounds awkward as it adopts the very technical medical terminology deemed necessary in the guiding principles report. In order not to diverge from the report, we prefer to retain the manuscript in its current style. However, we have now added full titles alongside all abbreviations, the first time each abbreviation is cited in the text.

  1. Discussions need to be substantially improved and detailed highlighting existing controversies, especially those related to differences in legislative regulations in various countries.

To our knowledge, only two reports (EMCDDA, 2003 and Institut de droit de la santé, 2012) have addressed comparative law between regulations in the European context and both are cited within the manuscript and set forth in the guiding principles (Chapter 1). We have not identified any other work on this subject.

  1. For documentation, the authors used a small number of bibliographic references and recent information in the field is missing.

In this viewpoint article we include only those references most relevant to the expert group`s perspective. These were the latest and most important references available to the expert group during their work on the guiding principles. In response to your suggestion, we have added another highly relevant article published shortly after this work was carried out (Degenhardt et al., 2019).

Thank you, once again for the time that you have taken to review this manuscript and provide your valued feedback.

Round 2

Reviewer 2 Report

Reviewer thanks authors for their thoughtful and candid responses. Reviewer has no further suggestions.

Reviewer 3 Report

The manuscript has been sufficiently improved to warrant publication in IJERPH.